# PTCHD1 Binds Cholesterol but Not Sonic Hedgehog, Suggesting a Distinct Cellular Function

**DOI:** 10.3390/ijms24032682

**Published:** 2023-01-31

**Authors:** Mimmu K. Hiltunen, Alex J. Timmis, Maren Thomsen, Danai S. Gkotsi, Hideo Iwaï, Orquidea M. Ribeiro, Adrian Goldman, Natalia A. Riobo-Del Galdo

**Affiliations:** 1Faculty of Biological and Environmental Sciences, University of Helsinki, 00100 Helsinki, Finland; 2School of Molecular and Cellular Biology, University of Leeds, Leeds LS2 9JT, UK; 3Astbury Center for Structural Molecular Biology, University of Leeds, Leeds LS2 9JT, UK; 4Institute of Biotechnology, HiLiFE, University of Helsinki, 00100 Helsinki, Finland; 5Leeds Institute of Medical Research, University of Leeds, Leeds LS2 9JT, UK

**Keywords:** Patched domain-containing 1, cholesterol, autism spectrum disorder, neurodevelopment, RNA granule

## Abstract

Deleterious mutations in the X-linked Patched domain-containing 1 (PTCHD1) gene may account for up to 1% of autism cases. Despite this, the PTCHD1 protein remains poorly understood. Structural similarities to Patched family proteins point to a role in sterol transport, but this hypothesis has not been verified experimentally. Additionally, PTCHD1 has been suggested to be involved in Hedgehog signalling, but thus far, the experimental results have been conflicting. To enable a variety of biochemical and structural experiments, we developed a method for expressing PTCHD1 in *Spodoptera frugiperda* cells, solubilising it in glycol-diosgenin, and purifying it to homogeneity. In vitro and in silico experiments show that PTCHD1 function is not interchangeable with Patched 1 (PTCH1) in canonical Hedgehog signalling, since it does not repress Smoothened in *Ptch1*^−/−^ mouse embryonic fibroblasts and does not bind Sonic Hedgehog. However, we found that PTCHD1 binds cholesterol similarly to PTCH1. Furthermore, we identified 13 PTCHD1-specific protein interactors through co-immunoprecipitation and demonstrated a link to cell stress responses and RNA stress granule formation. Thus, our results support the notion that despite structural similarities to other Patched family proteins, PTCHD1 may have a distinct cellular function.

## 1. Introduction

Autism (MIM 209850) is a neurodevelopmental condition with heterogeneous aetiology, characterised by restricted interests, repetitive behaviour, and atypical social interaction. Many genes have been associated with autism, several of which encode for proteins involved in synaptic processes, with mutations in the X-linked Patched domain-containing 1 (*PTCHD1*) gene (MIM:300828; Xp22.11) account for up to 1% of cases of autism [1,2,3,4]. *PTCHD1* is expressed throughout the brain and other tissues, and its expression profile in the brain changes as development proceeds [5,6,7]. The PTCHD1 protein is synaptically localised in neurons, and mutations disrupt its expression, stability, glycosylation, and localisation [7,8]. While the exact functional role of PTCHD1 remains elusive, several findings implicate an essential role in developmental processes and in the development of autism and intellectual disability [4,9]. 

Since the initial discovery in 2008, nearly 70 PTCHD1 copy number variations, microdeletions, or single-nucleotide variants have been linked to autism [5,8,10,11]. Phenotypically, rare genomic PTCHD1 variants are associated with minor facial dysmorphic features, autistic traits, and global developmental delay [9,12]. 

PTCHD1 is a member of the Patched domain-containing protein family, which also includes the Hedgehog (Hh) pathway receptor Patched 1 (PTCH1) and Niemann–Pick disease type C1 (NPC1). These proteins are structurally homologous and contain 12–13 transmembrane helices (TM), a sterol sensing domain (SSD) composed of five TMs, a sterol sensing-like domain, and two ectodomains (ECDs). Full-length PTCHD1 is an 888-amino acid, 12-pass transmembrane protein containing an SSD composed of TM 2-6 and a sterol sensing-like domain consisting of TM7-12 followed by a C-terminal tail containing a PDZ-binding motif (ITTV). Additionally, two large ECDs are inserted between TM1 and 2 and TM7 and 8.

The fact that both PTCH1 and NCP1 are involved in cholesterol transport raises the possibility that PTCHD1 also has cholesterol binding and/or transport activity. However, despite domain conservation, PTCH1 and NPC1 have very different functions. PTCH1 is both a receptor and a repressor of the Hh signalling pathway, which is initiated by the binding of a Hh ligand, such as Sonic Hedgehog (Shh) to the ECDs of PTCH1. Apo-PTCH1 inhibits the activity of the 7TM protein Smoothened by transporting cholesterol away from the outer leaflet of the plasma membrane, thus controlling the amount of available cholesterol. Shh binding blocks the transport activity of PTCH1, which increases cholesterol accessibility and allows Smoothened to transmit the Hh signal across the membrane to activate the GLI family of transcription factors [13,14,15]. Whether or not PTCHD1 could function as a Hh pathway modulator remains controversial: transient overexpression of PTCHD1 in 10T1/2 mouse fibroblasts reduced GLI1 expression; however, it did not reduce GLI-luciferase activity when expressed in *Ptch1*^−/−^ mouse embryonic fibroblasts (MEFs) [5,7,16]. This is not surprising since another close homolog to PTCH1, NPC1, has a different subcellular localisation and function. NPC1 mediates cholesterol egress from the late endosome and lysosomes in concert with NPC2 [17,18]. Mutations in *NPC1* or *NPC2* cause Niemann–Pick disease type C, characterised by the accumulation of cholesterol and sphingolipids in late endosomes and lysosomes, which results in developmental impairment and neurodegeneration [19,20].

As described above, cholesterol transporters are involved in fundamental cellular processes [21]. Understanding if PTCHD1 could have a similar function could provide key information on the association of PTCHD1 with autism. Unfortunately, very little is known about PTCHD1, and no structural analyses have been performed yet. Because many biochemical and structural experiments require purified, homogeneous protein, we have optimised the expression and purification of PTCHD1 from baculovirus-infected *Spodoptera frugiperda* (Sf9) cells and show that purified PTCHD1 is stable and homogeneous through size-exclusion chromatography multi-angle light scattering (SEC-MALS) analysis and negative staining electron microscopy (EM). We also gained insight into the unique features of PTCHD1 by combining in vitro and in silico methods, using an AlphaFold2-generated structure. Our results from a click-cholesterol assay indicate PTCHD1 can bind cholesterol in vitro, and in silico docking and comparison to PTCH1 supports this notion. Furthermore, through in silico prediction of complex formation, we find PTCHD1 ECDs do not bind Shh, unlike PTCH1 ECDs, suggesting the role of PTCH1 as a canonical Hh inhibitor is not shared by PTCHD1. Instead, a co-immunoprecipitation assay and subsequent identification and analysis of 13 PTCHD1-specific interactors seem to suggest a role in the cell stress response and RNA granule formation.

## 2. Results

### 2.1. Expression and Purification of PTCHD1

#### 2.1.1. Cloning, Expression, and Membrane Preparation

To enable purification and sterol binding experiments, we cloned human *PTCHD1* in a pFastBac^TM^ dual vector expression system to produce baculovirus-infected insect cells (BIICs) containing pFastBac^TM^-*PTCHD1*-CVGH vector with PTCHD1 fused to a C-terminal green fluorescent protein (GFP) with an 8× His tag and a human rhinovirus 3C (HRV3C) protease cleavage site to allow removal of the GFP-His tag (Figure 1A). The DH10Bac strain was deemed superior for BIIC production based on the number of positive colonies and yield of purified bacmid DNA. Pre-adapted Sf9 cells (1 × 10^6^ cells/mL) were infected with BIICs and harvested 48 h after proliferation arrest via centrifugation, yielding ~10 g cells/1 L culture. PTCHD1-GFP expression was confirmed indirectly by detecting the native yellow fluorescent protein fluorescence of the MultiBac^TM^ EmBacY plasmid (Figure 1B) and directly by detection of a ~100 kDa protein in the membrane fraction using GFP signal under blue light illumination following SDS-PAGE (GFP SDS-PAGE) (Figure 1C). As a membrane protein, PTCHD1-GFP was routinely observed to run faster than expected for a 125 kDa protein. Membranes were prepared by lysing the cells via sonication, followed by treatment with DNAse I and subsequent fractionation via centrifugation. Prepared membranes (1.2 g/1 L cell culture) were resuspended in 10 mM 4-(2-hydroxyethyl)-1-piperazineethanesulfonic acid (HEPES) pH 7.2, 200 mM NaCl, and 5% glycerol, flash-frozen, and stored at −80 °C. The resuspended, prepared membranes had an estimated total protein concentration of 20.6 mg/mL.

#### 2.1.2. Solubilisation Detergent Screening

Membrane protein purification requires a careful selection of detergents to extract proteins from membranes efficiently without causing aggregation or affecting stability. We first performed a small-scale solubilisation screen with several detergents using 2 g of flash-frozen membranes. We compared digitonin, glyco-diosgenin (GDN), lauryl maltose neopentyl glycol (LMNG), and N-dodecyl β-D-maltoside (DDM). LMNG and DDM were additionally supplemented with cholesteryl hemisuccinate (CHS) to mimic the presence of cholesterol. The membranes were incubated with the different detergents for 2 h, followed by SDS-PAGE analysis of soluble and membrane fractions to estimate the solubilisation yield. Densitometric analysis of the GFP fluorescence in SDS-PAGE gels (Figure 2A) revealed that 1.25% DDM or 1.25% LMNG supplemented with 0.125% CHS were the most efficient detergents, solubilising 73 and 65% of PTCHD1, respectively. For comparison, 0.2% digitonin allowed solubilisation of 41% of total PTCHD1, whereas 0.2% GDN solubilised only 27% of PTCHD1.

#### 2.1.3. Purification of PTCHD1 with GDN Produces a Monodispersed Peak by SEC-MALS

Despite the superior performance of LMNG+CHS in the PTCHD1-GFP solubilisation screen, SEC-MALS analysis of LMNG+CHS purification reveals a trace with several peaks and shoulders after the initial peak at ~7 min (Figure 2B). Therefore, we next tried GDN in subsequent purification trials of PTCHD1 because GDN was shown to aid protein stability, and both detergents have a low critical micelle concentration (CMC), which could offer several benefits for downstream applications such as cryo-EM. Following solubilisation, PTCHD1-HRV3C-GFP-His was purified by binding to HisPur^TM^ Cobalt resin, followed by stringent washes with 20 mM imidazole and eluted by in-column digestion with HRV3C protease to release PTCHD1. SEC-MALS analysis of spin-concentrated, purified PTCHD1 revealed that GDN produces a monodispersed peak at ~7 min, even after a round of flash freezing and storage (Figure 2C), suggesting good quality protein stabilised by GDN. The SEC-MALS peak at ~7 min was confirmed to be PTCHD1 by mass spectrometry (see next section). Therefore, despite initially obtaining better solubilisation yields with LMNG+CHS, large-scale PTCHD1 purification was continued using GDN as the detergent.

#### 2.1.4. Optimised Large-Scale Purification of GDN-Solubilised PTCHD1

We further optimised PTCHD1 purification by systematically altering several factors, including detergent concentration, solubilisation time, and resin amount (Appendix A). We also tested different buffers with no improvement in the final yield. We found that increasing the solubilisation time from 2 h to 18 h resulted in a 1.2-fold increase in solubilised PTCHD1, translating to a 1.9-fold increase in purified PTCHD1. Furthermore, doubling GDN concentration from 0.2 to 0.4% led to a 1.3-fold increase in solubilised PTCHD1. However, because these improvements still left up to ~70% of total protein unsolubilised, we decided to double the resin volume from 2 to 4% and further increase the concentration of GDN to 0.8%. Doubling the resin volume led to a 1.5-fold increase in eluted PTCHD1, while increasing GDN concentration led to a 2.7-fold increase in eluted PTCHD1.

The final protocol involved the solubilisation of Sf9 membranes with 0.8% GDN for 18 h and immobilisation of the soluble fraction on 7.5% (V/V) HisPur^TM^ Cobalt resin. Following two rounds of washes, PTCHD1 was released with HRV3C protease and spin concentrated in 0.004% GDN. We obtained a high-quality, homogenous sample of PTCHD1. The optimised protocol increased the yield by 1.7-fold: we obtained ~84 μg PTCHD1 from 13.5 g of Sf9 membranes (250 μL elution concentrated to 336.7 ng/μL). A Coomassie-stained SDS-PAGE of the purification (Figure 3A) showed a strong band corresponding to the approximate 100 kDa size of PTCHD1 in the concentrated elution fraction. Densitometry analysis of a GFP SDS-PAGE showed that ~70% of PTCHD1-GFP was solubilised from the total membranes, and ~55% bound to the resin. Negative staining with 2% uranyl acetate confirmed the homogeneity of the sample (Figure 3B). The purified protein was identified as PTCHD1 by mass spectrometry (Figure 3C and Appendix A). 

### 2.2. PTCHD1 Binds Cholesterol but Displays No Inhibitory Activity in Canonical Hedgehog Signalling

#### 2.2.1. Alignment Reveals Conserved Cholesterol-Binding Elements and a Lack of Ion Flux Motifs

An AlphaFold2 prediction of PTCHD1 shows that it is structurally homologous to several cholesterol transporters belonging to the Patched domain-containing family, including Dispatched 1 (DISP1), NPC1, and PTCH1 (Figure 4A). A sequence alignment of PTCHD1 with DISP1, NPC1, and PTCH1 showed a lack of conservation of the resistance-nodulation-division (RND) motifs GxxxDD and GxxxD/E (Figure 4B). The motifs containing ion-coordinating residues are conserved in TM4 and TM10 of the Hh pathway components DISP1, PTCH1, and PTCH2 but not in NPC1. Sequence and structural alignments showed that PTCHD1 lacks the RND motifs and does not contain ion-coordinating residues in the corresponding positions (Figure 4B and Appendix A). However, sequence alignment revealed a higher degree of conservation in the SSD (17%, 18%, and 23% identity with DISP1, NPC1, and PTCH1, respectively) than in the entire sequences. Furthermore, a proline on TM4 within the SSD, which is crucial for function in PTCH1 and NPC1 (P504 and P692, respectively), is conserved in PTCHD1 (P333) [22,23]. A superposition of PTCH1 (6DMY) and PTCHD1 (AF_Q96NR3) SSDs showed that P333 is in a near-identical position as P504 in PTCH1 (Figure 4B).

#### 2.2.2. Click-Cholesterol Assay Shows PTCHD1 Binds Cholesterol

Due to structural similarities to cholesterol transporters and P333 conservation in TM4 of the SSD domain, we investigated the functional relationship between PTCHD1 and cholesterol by performing a cholesterol click-reaction assay on purified PTCHD1 elution fractions. In brief, purified PTCHD1 was cross-linked with a PhotoClick cholesterol with UV light (λex = 365 nm). Afterwards, tetramethylrhodamine 5-carboxamido-(6-azidohexanyl) (5-TAMRA-Azide) was used to label cross-linked cholesterol by a ‘click-reaction’, followed by analysis by SDS-PAGE and imaging of 5-TAMRA-Azide (λex = 546 nm). The results showed that PTCHD1 can bind cholesterol (Figure 4C): strong bands of ~130 kDa and ~100 kDa, corresponding to the approximate molecular weights of fully glycosylated and unglycosylated PTCHD1 were present in all lanes of treated PTCHD1, solubilised in varying amounts of GDN. No bands are seen when cholesterol was omitted (‘no sterol’), indicating that 5-TAMRA-azide does not bind to PTCHD1 non-specifically in the absence of cholesterol. Additional negative controls confirmed specific binding of cholesterol to PTCHD1 (‘no UV radiation’, ‘no click-reaction’, and ‘no protein’).

#### 2.2.3. Docking Indicates PTCHD1 Can Bind Cholesterol Similarly to PTCH1

We superimposed PTCHD1 onto a PTCH1 structure and found that the SSDs are nearly identical in structure (Figure 4B). To further explore the similarities in cholesterol binding between PTCHD1 and PTCH1, we performed docking analysis using AutoDock Vina [22,23]. The full SSD of PTCHD1 (TM2-6 of AF-Q96NR3) or PTCH1 (TM2-6 of AF-Q13635), used as a control to validate the docking method, was defined as the docking site. We compared cholesterol binding by aligning docked cholesterols to the cryo-EM structure of PTCH1 (6RMG) with cholesterol-binding sites as defined by Qi et al. (2018): site 1 is within the outer leaflet pocket, site 5 is in the inner leaflet portion of the SSD, and site 4 is on the opposite side of the SSD, adjacent to TM1 (Figure 5B). 

Cholesterol docked to three sites on the PTCHD1 SSD with estimated binding energies ranging from −8.7 kcal/mol to −7.2 kcal/mol. The predicted cholesterol-binding positions in PTCHD1 directly overlapped with cholesterols in the PTCH1 cryo-EM structure: 25/27 cholesterol binding positions were in site 1 between TM3, TM4, and TM6. Additionally, two cholesterol binding positions were near site 5 in the putative inner leaflet portion of the SSD, adjacent to TM3 (Figure 5B and Appendix A). In the case of PTCH1-SSD, the estimated binding energy was between −8.4 kcal/mol and −7.3 kcal/mol. In addition, 23 docked cholesterol-binding positions overlapped with binding site 1 and four near site 4 (Figure 5B and Appendix A). The similar binding of cholesterol is unsurprising due to the SSDs of PTCHD1 and PTCH1 sharing nearly 23% identity (Figure 5C).

#### 2.2.4. PTCHD1 Does Not Inhibit Canonical Hedgehog Signalling

The structurally similar protein PTCH1 uses cholesterol to modulate Smoothened activity in canonical Hh signalling. However, whether PTCHD1 shares this function is controversial [5,7,16], so we decided to investigate the capacity of PTCHD1 to inhibit Hh signalling by conducting a GLI-luciferase assay in *Ptch1*^−/−^ MEFs, which display high constitutive GLI transcriptional activity due to the lack of endogenous PTCH1 (murine PTCH1 homologue). Our results indicated that transient expression of PTCHD1 does not reduce GLI-luciferase activity compared to the empty plasmid (pcDNA3.1^+^ control), whereas transfecting cells with PTCH1 inhibited GLI-luciferase activity by ~65% (Figure 6A).

#### 2.2.5. Structural Analysis and Complex Prediction Showed That PTCHD1 Is Unlikely to Bind Shh

We next investigated the possibility that PTCHD1 could serve as a decoy receptor for Shh. We used AlphaFold2 to predict the complex formation of N-terminal Shh (ShhN, res 23-197) with the ECDs of PTCHD1 and, as a positive control, the ECDs of PTCH1. Over 200 sequences were available for multiple sequence alignments in all positions of ShhN and each ECD. Furthermore, the local distance difference test (lDDT) scores were above 50 for most positions, excluding the first residues of ShhN, either ECD1 or the last residues of ECD2 (Appendix A). Alignment of the predicted PTCHD1 ECD1 and ECD2 onto the full AlphaFold2 predicted structure (AF-Q96NR3) indicates the conformation of each ECD and their interaction has been predicted well, despite running each ECD separately (RMSD 0.961 Å). ShhN was predicted to bind to a β-hairpin and an α-helix on ECD2 of PTCHD1 (Figure 6C, left). However, the predicted aligned error was very high between ShhN and the ECDs for PTCHD1 (Figure 6B, left, Appendix A), suggesting ShhN is unlikely to form a complex with PTCHD1 ECDs. Conversely, in the predicted PTCH1:Shh complex, ShhN sits between two loops of ECD1 and a loop on ECD2, and the predicted aligned error was low throughout (Figure 6B, right, Appendix A). Furthermore, the predicted PTCH1:Shh structure aligned well with a complex solved by cryo-EM (RMSD 1.606Å with 6DMY) and contains 6/10 of the hydrogen bonds found in the complex (Figure 6C, right, Section 2.2.6), indicating that AlphaFold2 can predict Shh binding.

Finally, a LigandTracer^TM^ experiment with tethered Sf9 cells expressing PTCHD1-GFP with 2–800 nM labelled ShhN showed no binding between the two within 60 min (Figure 7). PTCHD1-GFP expression in Sf9 cells tethered to a Petri dish was confirmed via detecting YFP and GFP fluorescence directly, and successful ShhN labelling with DyLight^TM^ 650 was confirmed by absorbance measurements (Figure 7A,B). In the LigandTracer^TM^ measurement (Figure 7C), no signal was detected between 15–25 and 28–38 min with 2 and 20 nM ShhN, respectively. Upon increasing the concentration of ShhN to 200 nM, a sharp increase in the signal was detected at ~40 min in one of the triplicates. However, the signal reverted to the baseline within 2 min. After increasing ShhN concentration to 800 nM, there was an increase in the background signal. These changes in signal intensity were thought to be due to nonspecific interaction of ShhN with the cell membrane, due to the hydrophobicity of ShhN, as ligand binding typically results in a permanent increase in signal intensity in test cells but a very low, temporary increase in control cells [24].

#### 2.2.6. Structural Analysis Reveals Key Differences in PTCHD1 and PTCH1 Ectodomains

Through further structural analysis, we identified critical differences between PTCHD1 and PTCH1, which may help explain why PTCHD1 is unlikely to bind Shh. Residues in the PTCH1–Shh interface are mainly polar or charged [25] and PISA analysis of a PTCH1–Shh complex structure (6DMY) indicates hydrogen-bonds formed by ten PTCH1 residues (E212, D217, E221, Y222, G378, Y379, E380, E947, D951, and E958) mediate the interaction between PTCH1 and Shh. The hydrogen-bonding residues are in ECD1 loop 1 (res 206–213), ECD1 loop 2 (res 376–388), and an ECD2 loop (res 943–969). Sequence alignment of PTCH1 and PTCHD1 ECDs revealed an overall identity of ~11% and only two of the ten hydrogen-bonding residues of PTCH1 were conserved in PTCHD1 (Appendix A). Superposing the PTCHD1 structure onto PTCH1 clearly illustrated the difference between the ECDs; the secondary structure and charge distribution of the two have little in common. The three PTCH1 loops primarily responsible for Shh binding, composed of 24, 13, and 32 residues, form a well-defined negatively charged binding pocket. In contrast to PTCH1, there is no well-defined negatively charged binding pocket on the PTCHD1 surface, only one loop corresponding to ECD1 loop 1 of PTCH1, and a β-hairpin replaces the ECD2 loop (Figure 6C). Finally, the PTCHD1 ECD1 loop consists of only 16 residues and the ECD2 β-hairpin of 23 residues. The absence of a negatively charged binding pocket with flexible loops containing hydrogen-bonding residues, along with the high predicted aligned error of the complex formation, strongly suggest that PTCHD1 does not bind Shh and, thus, could not act as a decoy Shh receptor. 

### 2.3. PTCHD1 Specific Interaction Network Linked to Ribonucleoprotein Granule Proteins and Stress Granules

We performed a co-immunoprecipitation assay of PTCHD1 in HEK293 cells, followed by identifying protein–protein interactions from the lysate by mass spectrometry. Ninety-five proteins, with a 10 lgP score above a set threshold and ID’d with at least three unique peptides, were found to co-immunoprecipitate with PTCHD1 (Appendix A). To identify PTCHD1-specific interactors, we compared the hits from mass spectrometry against those immunoprecipitated by the highly similar proteins PTCH1 and PTCH2. After removing common hits, we identified 13 PTCHD1-specific protein–protein interactors (Table 1). Remarkably, two of the PTCHD1-specific interactors are independently associated with autism (DYRK1A and DDX3X) and 11 have been previously identified as PTCHD1 interactors in mouse brain lysates [6]. Therefore, our results provide independent confirmation of 11 PTCHD1-interacting proteins.

#### Gene Ontology Analysis 

We employed gene ontology (GO) analysis to identify protein classes and molecular functions among the PTCHD1-specific interactors. ATXN2L, PABP1, G3BP2, and DDX6 belong to RNA metabolic proteins, while DYRK1A and UBP34 are protein-modifying enzymes. CSPR1 and HSPA8 are cytoskeletal and chaperone proteins, respectively. The protein class of four interactors, NUFIP2, UBB, ATAD3A, and DCAF7, is unidentified. According to the GO analysis, the molecular function of ATXN2L, NUFIP2, PABP1, UBB, CSRP1, G3BP2, ATAD3A, and DDX6 is binding, while DYRK1A and UBP34 are involved in catalytic activity. DYRK1A is additionally involved in transcription regulation. Both DDX3X and HSPA8 fall into three functional classes: binding, catalytic activity, and ATP-dependent activity. Statistical over-representation tests reveal significantly over-represented (*p* < 0.01, false discovery rate < 0.01) biological processes, protein classes, and molecular functions compared to the entire *Homo sapiens* genome used as a reference (Table 2). Notably, RNA-binding proteins involved in stress granule (SG) and ribonucleoprotein granule (RNP) assembly were over-represented within the PTCHD1-specific interactors, which suggests a role for PTCHD1 in these processes.

Finally, we performed a STRING network analysis of the 13 PTCHD1-specific interactors identified here. The analysis showed that six of them (ATXN2L, DDX3X, DDX6, G3BP2, NUFIP2, and PABP1) form a network of experimentally determined physical protein–protein interactions (Figure 8). Notably, all six interactors are associated with RNA granules (Table 1). This tight network suggests that PTCHD1 could be associated with stress granules. In support of this possibility, we found that overexpressed PTCHD1-eGFP colocalizes in perinuclear structures positive for the stress granule marker G3BP (Figure 9). Formation of low-level stress granules was observed in cells transfected with empty or PTCHD1-encoding plasmids using Lipofectamine2000, under otherwise non-stressed conditions. DYRK1A and DCAF7 also physically interact, while the final five (UBB, UBP34, ATAD3A, HSPA8, and CSRP1) are not part of known physical interactions.

## 3. Discussion

In this study, we report for the first time the purification to homogeneity of PTCHD1, a member of the Patched domain-containing family implicated in autism, in the presence of a sterol analogue. Using the purified protein, we demonstrated that PTCHD1 can directly bind cholesterol, a potential substrate suggested by the presence of an SSD motif within TM2-6. Given the structural similarity and the fact that PTCH1, a Hh receptor, also binds and transports cholesterol [15,28], it had been proposed that PTCHD1 could serve as an alternate Hh receptor. However, experimental evidence in two different studies came to contradictory conclusions. A combination of in vitro and in silico methods led us to conclude that PTCHD1 and PTCH1 have divergent biological functions. More precisely, while PTCH1 is a Shh receptor and regulates the canonical Hh pathway, PTCHD1 cannot bind Shh and is incapable of rescuing *Ptch1* deficiency in cells. 

High-resolution structures of the Patched domain-containing proteins PTCH1 and NPC1 have been previously solved [28,29]. As we also wish to obtain a structure, we designed a similar approach for purification of PTCHD1 from baculovirus-infected Sf9 cells. Since PTCHD1 contains an SSD, we included CHS, a sterol analogue, that could stabilise it during purification. After screening different detergent combinations for solubilisation, GDN proved to offer the best compromise of yield and homogeneity, as determined by mass spectrometry analysis, SEC-MALS, and negative staining EM of the purified PTCHD1. While the yield of purified PTCHD1 was relatively low, it is sufficient for many functional assays and structural methods, including cryo-EM.

Next, we combined in vitro click chemistry experiments with in silico docking, sequence, and structural analysis to explore cholesterol binding to PTCHD1. Our in vitro data confirmed that PTCHD1 binds cholesterol (Figure 4C), and docking analysis suggested that the binding positions overlap with previously identified positions in PTCH1 [28] (Figure 5B and Appendix A). While the chosen docking protocol may not be optimal for accurate prediction of ligand binding in a membrane environment, the predicted cholesterol-binding positions were close to experimentally identified sites, and the predicted binding energy was similar between PTCHD1 and PTCH1, suggesting that PTCHD1 binds cholesterol similarly to PTCH1. However, it remains unclear whether cholesterol is a relevant endogenous cargo. 

Recently the structure of another close relative, DISP1, a Hh pathway activator, was shown to contain Na^+^ ions coordinated by three charge-neutralising aspartates at the interface of TM4:TM10 and TM5:TM11 [30]. The residues on the TM4:TM10 interface are part of the conserved RND permease family motifs GxxxDD and GxxxD/E, essential for coupling ion flux to pump small hydrophobic compounds against a chemical gradient. Wang et al. (2021) proposed that DISP1 uses Na^+^ flux in powering its activity and further speculated that the functional role of PTCH1 involves ion flux since many of the 13 identified Na^+^ associated residues are conserved. Mutating the residues impairs the function of PTCH1 and DISP1, and a Na^+^ or K^+^ gradient is required for PTCH1 function [30,31,32]. Neither RND motif is conserved in PTCHD1, while only GxxxD/E is conserved in NPC1 (Figure 4B). Due to the lack of charge-neutralising residues in both, and NPC1 being known to transport cholesterol along its concentration gradient [33], we propose that PTCHD1 could transport cholesterol along its concentration gradient too. 

In contrast to findings reported by Noor et al. (2010) and Chung et al. (2014) [5,16], but in agreement with Ung et al. (2018) [7], our results suggest PTCHD1 does not inhibit canonical Hh signalling like PTCH1 (Figure 6A). Further, Ung et al. (2018) [7] hypothesized that PTCHD1 was unlikely to bind Shh due to the lack of residues corresponding to residues ^902^LTKQRLVDADG^912^ in PTCH1. While it is true that PTCHD1 does not possess these residues, this fact is unlikely to contribute to the inability to bind Shh. Monomeric, dimeric, and tetrameric PTCH1:Shh structures (e.g., 6RMG, 6E1H, and 6N7K) showed that Shh is only near these in the tetrameric structure (6N7K), and even then, an analysis of the interface showed they are not main contributors for binding Shh. However, it is possible residues ^902^LTKQRLVDADG^912^ influence Shh binding transiently. 

AlphaFold2 has been shown to predict heterodimeric protein complexes and accurately separate interacting proteins from non-interacting proteins [34]. We used it to predict the complex formation of Shh with the PTCHD1 ECDs and found a low likelihood of complex formation between them (Figure 6B, left), which is consistent with the LigandTracer^TM^ experiment showing there is no binding of Shh to PTCHD1 expressed in Sf9 cells. The structural analysis suggests the lack of binding is due to the absence of ECD loops, which form a negatively charged pocket and mediate binding in PTCH1 (Figure 6C). The inability of PTCHD1 to bind Shh and to inhibit GLI transcription in MEFs argues against the possibility of acting as a Hh receptor, despite sharing structural features with PTCH1. The absence of Hh-related phenotypes in PTCHD1-deficient mice also supports our conclusion [4,6,7]. Although our in vitro experiments and in silico structural analysis suggest that PTCHD1 does not bind Shh nor inhibit Hh signalling, we cannot rule out that PTCHD1 influences the Hh pathway in a noncanonical fashion and/or in a cell type-specific manner.

Co-immunoprecipitation in HEK293 cells followed by MS and GO analysis identified 13 PTCHD1-specific protein interactors that do not associate with PTCH1 nor PTCH2. While our study was limited to HEK293 cells, given the difficulty of detecting PTCHD1 expressed in a neuronal cell line, 11 of the 13 PTCHD1-specific interactors have also been previously identified as PTCHD1 interaction partners using adult male mouse brain lysates (Table 1) [6], giving confidence in our approach. Interestingly, these interactors are significantly enriched in proteins involved in cellular stress responses, RNP assembly, and other associated processes (Table 1 and Table 2). SGs are RNP cytoplasmic aggregates of untranslated mRNAs bound to RNA binding proteins [35]. These membrane-less organelles are involved in RNA storage and decay and regulation of local translation [36,37]. RNA-binding protein dysfunction and SG or RNP formation have been linked to neurodevelopment and neurodegeneration [36,38]. For example, mutations in DDX3X, one of the PTCHD1-specific interactors (Table 1), disrupt RNA metabolism, induce neuronal RNA granule formation, and lead to intellectual disability [39]. Importantly, a GO enrichment analysis of the *Ptchd1*^-/Y^ mouse brain hippocampal transcriptome shows many of the same GO processes are enriched in down-regulated genes: cellular stress response (GO: 0033554 and 0006950), RNA-binding (GO: and 0044822), non-membrane bounded organelles (GO: 0043228 and 0043232), and ribonucleoprotein complexes (GO:0030529) [7]. These findings and our study lend credence to the idea that PTCHD1 is involved in RNA granule assembly. 

RNA granule formation is also associated with Fragile X syndrome, one of the most common single-gene causes of autism: the protein underlying Fragile X syndrome, FMRP, localises in neuronal RNA granules, regulates their formation, and regulates local translation independently and in complex with other proteins [40,41]. Interestingly, the three PTCHD1 interactors with the highest number of unique peptides (i.e., ATXN2L, NUFIP2, and PABP1) and all interactors forming a cluster of physical interactions (Figure 8: ATXN2L, DDX3X, DDX6, G3BP2, NUFIP2, and PABPC1) are also known to interact with Fragile X mental retardation protein (FMRP) [42,43,44]. It is thus tempting to speculate that there could be a convergence in the cellular role of PTCHD1 and FMRP. Studying this new and unexpected link will be the focus of our future research. 

## 4. Materials and Methods

### 4.1. Cloning and Production of PTCHD1 Baculovirus-Infected Cells

Wild-type PTCHD1 cDNA (RefSeq NG_021300) was In-Fusion^TM^ cloned into a pOPINN vector and transformed into DH5alpha. Positive colonies were selected after O/N culture on Luria Broth (LB) agar plates with ampicillin. The colonies were grown for a further 8 h at 37 °C, 225 rpm in LB with ampicillin before purifying the plasmid using the EZ-10 Spin Column Plasmid DNA Miniprep Kit (Bio Basic, Markham, ON, Canada) according to the manufacturer’s instructions. Successful cloning was confirmed by Sanger sequencing.

A pFastBac^TM^ vector expression system was then used to produce baculovirus-infected Sf9 cells. *PTCHD1-HRV3C-GFP-H6* was PCR-amplified from the pOPINN plasmid, restriction digested and cloned into a pFastBac^TM^-CVHG vector. To determine an optimal expression strain, the generated bacmid construct was transformed into two *E. coli* strains, DH10Bac (Invitrogen, Waltham, MA, USA) and MultiBac^TM^ EmBacY (Geneva Biotech, Geneva, Switzerland) [45] and grown O/N at 37 °C, 225 rpm in Luria Broth and subsequently plated on selective LB plates and incubated 24 h at 37 °C. Positive colonies were identified via blue/white screening, expanded, and the bacmid DNA was purified using the EZ-10 Spin Column Plasmid DNA Miniprep Kit (Bio Basic). Successful cloning was confirmed by Sanger sequencing.

### 4.2. Expression in Sf9 Insect Cells and Membrane Preparation

PTCHD1 was expressed in Sf9 cells maintained at a density of 1–2 × 10^6^ cells/mL in Insect-XPRESS^TM^ Protein-free Insect Cell Medium with L-Glutamine (Lonza, Basel, Switzerland) at 27 °C, 5% CO_2_, 120 rpm. After the cells doubled every 24 h, BIICs were thawed and added 1:1500 *v*/*v* to the pre-adapted Sf9 cells. Successful transfection was confirmed indirectly by observing the fluorescence of the native yellow fluorescent protein within the Multibac^TM^ EmBacY plasmid. The cells were harvested 48 h after infection via centrifugation (800× *g* for 25 m), resuspended in 10 mM HEPES, 200 mM NaCl, and 5% glycerol, and pelleted again by centrifugation (900× *g* for 15 m). The supernatant was subsequently removed, and the pellets were flash frozen in liquid nitrogen and stored at −80 °C.

Membranes were prepared via sonication at 40% amplitude for 2 min (10 s on, 5 s off) in 10 mM HEPES, 200 mM NaCl, 2.5 mM MgCl_2_, 0.5 mM CaCl_2_, supplemented with 1 × Proteoloc^TM^ and 10 mM phenylmethylsulphonyl fluoride protease inhibitors. The lysate was pelleted via centrifugation, and the membranes were resuspended in 10 mM HEPES, 200 mM NaCl, and 5% glycerol. The membranes were flash frozen in LN_2_ for storage at −80 °C. PTCHD1 expression and membrane fractionation were confirmed with SDS-PAGE, initially visualising PTCHD1-GFP using blue light and then by Coomassie blue staining. The samples were prepared in 4 X buffer and ~50 μg protein was loaded into each well.

### 4.3. Small-Scale Detergent Screening 

Small-scale solubilisation tests were conducted with four detergent candidates on 2 g of prepared PTCHD1 membranes to find a suitable detergent. Four detergents were tested: 0.2% digitonin or GDN and 1.25% LMNG or 1.25% DDM, both supplemented with 0.125% CHS. All small-scale solubilisation tests were performed at 4 °C for 2 h in 1 × PBS, 200 mM NaCl, and 10% glycerol. The results were assessed by SDS-PAGE.

### 4.4. Optimisation of PTCHD1 Purification

All steps were performed at 4 °C unless otherwise specified. To solubilise PTCHD1, ~13 g of membranes was incubated in 1.25% LMNG + 0.125% CHS or 0.2−0.8% GDN for 2−18 h in 10 mM HEPES, 200 mM NaCl. The insoluble fraction was pelleted via centrifugation (42,000× *g* RPM, 1h), and the solubilised supernatant was incubated with 2–7.5% (V/V) HisPur^TM^ Cobalt resin, rotating end over end for 1.5 h. The resin was sedimented via centrifugation (1000× *g* for 3 m), and unbound supernatant was removed. The resin was resuspended in 30 column volumes (CV) equilibration buffer (10 mM HEPES, 200 mM NaCl, 0.004% GDN), transferred to an Econo-Pac® Chromatography Column (BIO-RAD), and subsequently washed with 20 CV 10 mM HEPES, 200 mM NaCl, 20 mM imidazole, and 0.004% GDN. The column was washed a second time with 20 CV equilibration buffer to remove residual imidazole. The resin was resuspended in 1 CV equilibration buffer and incubated for 1 h with 1:5 human rhinovirus 3 C protease to release PTCHD1. The resin was sedimented via centrifugation (1000× *g* for 3 min), and the supernatant was collected. The resin was resuspended in 0.5 CV equilibration buffer, sedimented, and the supernatants pooled. The protein concentration was estimated by measuring the absorbance at 280 nm, and the sample was spin-concentrated 5−8-fold using a 30 kDa molecular weight cut off Vivaspin 500 polyethersulfone membrane centrifuge concentrator (Sartorius, Goettingen, Germany) at 4000× *g* rpm in 3 min intervals with frequent pipetting to prevent aggregation. 

The purification results were first assessed by SDS-PAGE. Spin-concentrated PTCHD1 elution fractions for both 1.25% LMNG + 0.125% CHS and 0.2% GDN were then subjected to SEC-MALS analysis. The predominant product of GDN purification was subjected to mass spectrometry analysis to confirm the presence of PTCHD1. Homogeneity and distribution of the sample were assessed via negative stain EM. The sample was diluted 1:5 before staining with 2% uranyl acetate at RT and imaging with an FEI Technai F20, 200 KeV, equipped with a FEG electron source and an FEI CETA (CMOS CCD) camera at the electron microscopy facility of the University of Leeds. Screen down and up nominal magnifications were 26,500× and 30,000×, respectively, with a pixel size of 0.351 nm. 

### 4.5. PhotoClickable Cholesterol Assay

Purified PTCHD1 was incubated for 30 min on ice with 6 μM inclusion bodies of PhotoClick cholesterol (β-methyl-cyclodextrin and hex-5’-ynyl 3β-hydroxy-6-diazirinyl-5α-cholan-24-oate, Avanti Polar Lipids). The samples were irradiated with UV light at 365 nm using a CL-1000 Ultraviolet crosslinker (UVP, Upland, CA, USA) and incubated for 1 h at RT in click reagent solution (100 μM 5-TAMRA-Azide, 1 mM CuSO4, 1 mM tris(2-carboxyethyl)phosphine hydrochloride, and 100 μM tris(benzyltriazolylmethyl)amine). The reaction was stopped with 10 mM EDTA, and excess click reagents were precipitated with 4 x volume of ice-cold acetone before centrifugation for 10 min at 4 °C, 13,000× *g* RPM. The pellet was washed twice with cold MeOH, re-centrifuged, and air-dried before resuspending in 2 x Laemmli buffer. An SDS-PAGE gel was run, and the bands were visualised with a Cy3/TAMRA/Rhodamine filter set (Ex/Em ~540/568 nm). A no cholesterol reaction was used to ensure that 5-TAMRA-Azide does not bind non-specifically to PTCHD1 or the GDN micelles. Further controls omitting the purified protein, the UV radiation required for cross-linking, or excitation at 546 nm required for the click reaction were used to ensure the cholesterol probe specifically interacts with PTCHD1.

### 4.6. GLI Luciferase Assay in Mouse Embryonic Fibroblasts

*PTCH1*^−/−^ MEFs, a kind gift from Dr. Matthew Scott (Stanford University), were grown to 90% confluence and subsequently washed, detached, and diluted 1:5 in DMEM, 10% FBS, and 1% GlutaMAX^TM^ for seeding in 24-well culture plates. After 24 h, the cells were transfected with 7.5 ng/μL *PTCH1-HA*, *PTCHD1-His*, or pcDNA3.1^+^ plasmid as well as Firefly- and Renilla-luciferase reporter plasmids (p8xGBS-Luc and pRL-SV40) using TransIT-X2® (Mirus Bio LLC, Madison, WI, USA) in Opti-MEM^TM^ reduced serum media (Thermo Fisher Scientific, Waltham, MA, USA) and incubated for 24 h at 37 °C, 5% CO_2_. All conditions were performed as quadruplicates. Transfected cells were washed with PBS before replacing the media with DMEM, 0.5% FBS, and 1% GlutaMAX^TM^ for 48 h. After serum starvation, the cells were washed with PBS and lysed with 1 × passive lysis buffer at RT with shaking for 15 min. Firefly- and Renilla-luciferase activities were determined with the dual-luciferase reporter assay system (Promega, Madison, WI, USA) in a Promega GloMax® 20/20 luminometer. The measurements were obtained in relative luciferase units and expressed as a ratio of Firefly luciferase to Renilla luciferase. A Student’s *t*-test was conducted to reveal statistically significant differences. 

### 4.7. Sequence Analysis

To compare PTCHD1 to known sterol transporters (PTCH1, DISP1, and NPC1) sequence alignments were performed using Clustal Omega version 1.2.3 [46] and structure-based sequence alignments with PyMOL version 2.5.2 (Schrödinger, New York City, NY, USA).

### 4.8. ShhN-ECD1-ECD2 Complex Prediction

AlphaFold2, run through ColabFold [47,48], was used to assess the interaction of PTCHD1 and the N-terminal domain of ShhN (res 23-197) by predicting the complex formation of the PTCHD1 ECDs (res 51-252 and 526-688) and ShhN. As a control, complex formation of the PTCH1 ECDs (res 122-436 and 770-1027) with ShhN was predicted and compared with the results from the 6RMG cryoEM structure. The predictions were template-free and AMBER force fields were incorporated [49]. The local distance difference test and predicted aligned error scores were evaluated to assess predicted complexes [50]. Interaction interfaces were identified with the PDBe PISA server (version 1.52) [51]. The sequences were obtained from UniProt (PTHD1_Human Q96NR3, PTC1_Human Q13635, SHH_human Q15465).

### 4.9. LigandTracer^TM^ Experiment

Using a method previously described by Bondza et al. (2017) [24], we tethered Sf9 cells expressing PTCHD1-GFP to a Petri dish with a biomolecular anchor molecule (BAM, SUNBRIGHT® OE-040CS, NOF Corporation, Tokyo, Japan) and analysed ShhN binding with LigandTracer^TM^ (Ridgeview Instruments, Uppsala, Sweden). Briefly, ShhN was labelled with 10 μg of DyLight^TM^ 650 per 1 mg of protein in 50 mM Tris-HCl pH 8.0, 200 mM NaCl, 2 mM CaCl_2_, 0.1% Tween-20 by incubating at RT for 1 h in the dark. Four drops of the BAM (4 mg/mL in MQ) were pipetted to defined areas on a Petri dish and incubated at RT for 1 h prior to removing any excess BAM. Sf9 cells expressing pFastBac-PTCHD1-CVHG (see above) were added to three drops, and Sf9 cells without PTCHD1 expression to the fourth as a control. After 40 min incubation at RT, the excess media was removed, blocking media (Insect XPRESS^TM^ media supplemented with 5% bovine serum albumin) was added, and the dish was incubated at 27 °C O/N. Before starting the LigandTracer^TM^ measurements, the blocking media was replaced with fresh media supplemented with 1 mM CaCl_2_. A baseline measurement was collected for 15 min prior to adding 2 nM labelled ShhN. Measurements of the triplicate test areas along with the control area were stopped at 10 min intervals to increase the amount of ShhN to 20, 200, and finally 800 nM totalling 50 min of measurements with ShhN. The ShhN media was then replaced with fresh media without a ligand, and a dissociation curve was recorded for 30 min.

### 4.10. Docking Analysis

Docking analysis of cholesterol to the putative PTCHD1 structure (AF-Q96NR3) was performed with AutoDock Vina [22,23]. The protein was prepared via AutoDockTools [52] by removing solvent/ligand molecules, followed by adding polar hydrogens and Gasteiger charges. The search space was defined as the full SSD (TM2-6) and TM1, with 0.37 Å grid spacing, and docking analyses were run thrice with an exhaustiveness, i.e., number of independent runs, of 40 [22,23]. The SSD of PTCH1 (AF-Q13635) was used to validate the docking procedure. The structure was prepared as above. The positions of docked cholesterols were compared to the PTCH1 cryoEM structure (6RMG) via structural alignment on PyMOL version 2.5.2 (Schrödinger).

### 4.11. HEK293 Cell Culture and Co-immunoprecipitation Assay

HEK293 cells (American Type Culture Collection) were maintained in Dulbecco’s Modified Eagle’s Medium (Gibco^TM^, Thermo Fischer Scientific), supplemented with 10% foetal bovine serum and 1% GlutaMAX (Gibco^TM^, Thermo Fischer Scientific). The cells were grown at 37 °C, 5% CO_2_ and sub-cultured in 1/20 – 1/5 ratios prior to reaching confluence by washing the cells with PBS, detaching them with 25% trypsin-EDTA, and diluted in fresh media.

HEK293 cells (grown to >90% confluence in 10 cm culture dishes) were transiently co-transfected with 4 μg of each plasmid (Ptch2-FLAG or Ptchd1-CVGH) along with Lipofectamine^TM^ 2000 (Thermo Fischer Scientific). After 24–36 h incubation at 37 °C, 5% CO_2_ the transfected cells were harvested and lysed by rotating for 30 min end over end at 4 °C in lysis buffer (50 mM Trish-HCl pH 7.5, 150 mM NaCl, 1% Nonidet^TM^ P-40, 0.5% sodium deoxycholate, 0.4 mM phenylmethysulphonyl fluoride, 1 mM DTT). The lysate was pelleted via centrifugation at 13,000× *g* rpm for 15 min and the supernatant was removed. Subsequently, the lysate was incubated for 1.5 h at 4 °C with a primary epitope tag antibody (anti-FLAG or anti-His, Proteintech, Rosemont, IL, USA) before adding Dynabeads® (Thermo Fisher Scientific) and further incubating for 2 h. The beads were collected using a magnetic rack and washed three times with lysis buffer. The lysates were treated with 2 x Laemmli buffer and heated at 45 °C for 25 min before running an SDS-PAGE gel.

### 4.12. Mass Spectrometry

Purified protein band identification, protein post-translational modification mapping, and immunoprecipitation interaction protein identification was performed by the mass spectrometry facility at the University of Leeds. Co-immunoprecipitates and Coomassie-stained SDS-PAGE gels were delivered to the facility, where gel extraction and protein digestion was performed with trypsin, chymotrypsin, asp-N, and trypsin with lys-c. The data was processed with PEAKS (Bioinformatics Solutions Inc., Waterloo, ON, Canada).

### 4.13. Gene ontology Analysis

GO analysis was performed through the PANTHER database online server v. 17.0 [53,54]. Statistically significant, overrepresented GO classes were identified with the PANTHER Overrepresentation Fischer’s exact test (PANTHER v. 17.0, release 12 July 2022, available online http://www.pantherdb.org/tools/compareToRefList.jsp (accessed on 29 July 2022) with false discovery rate correction using the complete *Homo sapiens* genome (PANTHER v. 17.0, reference list 20589, release 22 March 2022, available online www.pantherdb.org (accessed on 29 July 2022)), accessed through the PANTHER database, as a reference [55]. 

### 4.14. Confocal Microscopy

HEK293 cells (3 x 10^6^/ml) were seeded onto round 13 mm coverslips and transiently transfected with empty vector (pcDNA3.1) or PTCHD1-eGFP using Lipofectamine 2000. Twenty-four hours post-transfection, the cells were fixed with 4% paraformaldehyde for 15 min, permeabilized with ice-cold methanol at −20 °C for 10 min, and blocked with 2% BSA in PBS for 1 h at RT. Following blocking, the primary antibody anti-G3BP (rabbit Ab, Abcam EPR13986(B)) was added at 1:200 dilution in PBS/2% BSA and incubated at 4 °C overnight. After 3 PBS washes, a goat anti-rabbit secondary antibody conjugated to AlexaFluor 568 (Invitrogen, A-11011) was added at 1:500 in PBS/2% BSA for 1 h at RT. The cells were washed and mounted on glass microscope slides with Fluoroshield with DAPI (Sigma, F6057). Confocal images were acquired using the Zeiss LSM880 upright confocal microscope at the UoL Bioimaging Facility using a 63 x oil objective for visualisation of stress granules. Images were analysed using ImageJ with Fiji.

## 5. Conclusions

In conclusion, our findings support the idea that PTCHD1 is functionally unique and participates in fundamental cellular processes that are different to those of PTCH1 and NPC1. First, we showed that PTCHD1 is not a PTCH1 functional homolog. Second, we showed that PTCHD1 binds cholesterol but not Shh and cannot inhibit canonical Hh signalling. Third, we identified key features in the PTCHD1 sequence and structure that distinguish it from other Patched domain-containing family members. Finally, we identified 13 PTCHD1-specific interactors, the functions of which suggest a link to stress granule formation. 

## Figures and Tables

**Figure 1 ijms-24-02682-f001:**
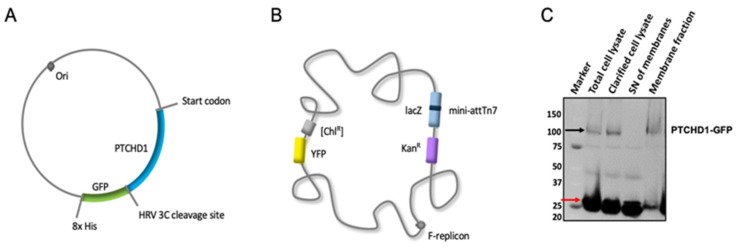
PTCHD1 expression and membrane fractionation. (**A**) A schematic representation of pFastBac-*PTCHD1*-CVGH. Ori stands for origin of replication. PTCHD1 is followed by an HRV 3C cleavage site, GFP, and 8× His tag. (**B**) A schematic representation of the MultiBac^TM^ EmBacY. (**C**) SDS-PAGE of PTCHD1-GFP expressed in Sf9 cells infected with BIICs shows clear bands around 100 kDa, corresponding to the approximate size of PTCHD1-GFP (black arrow), and around 25 kDa, corresponding to the native YFP of the MultiBac EmBacY bacmid (red arrow).

**Figure 2 ijms-24-02682-f002:**
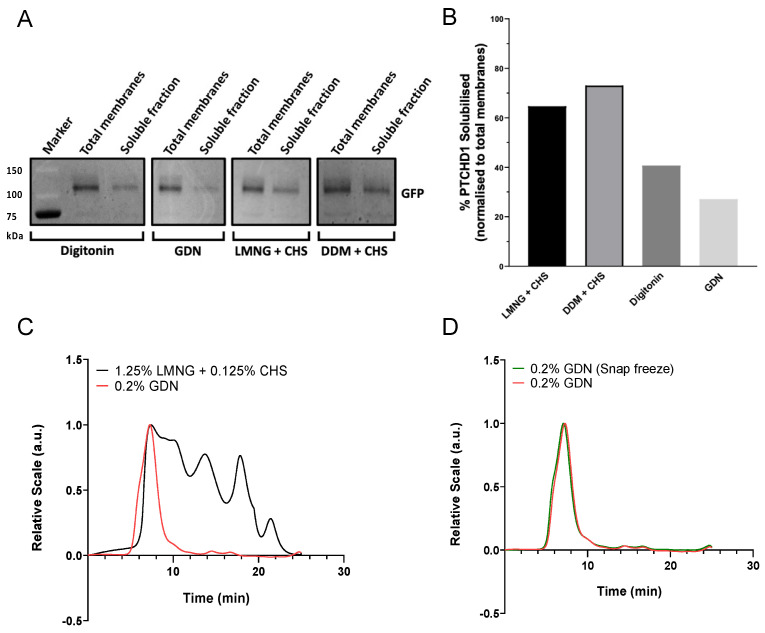
PTCHD1-GFP solubilisation and purification. (**A**,**B**) Small scale detergent screening indicates 1.25% DDM with 0.125% CHS solubilises PTCHD1-GFP from 2 g of membranes most efficiently (73%), closely followed by 1.25% LMNG with 0.125% CHS (65%). Solubilisation with 0.2% digitonin or GDN resulted in <40% solubilised PTCHD1. (**C**) SEC-MALS analysis of purified PTCHD1 from 13.5 g membranes solubilised in either 0.2% GDN or 1.25% LMNG + 0.125% CHS shows solubilisation in GDN results in a homogeneous sample, unlike solubilisation in LMNG. (**D**) The SEC-MALS trace of PTCHD1 purified in 0.2% GDN before and after snap freezing in liquid nitrogen.

**Figure 3 ijms-24-02682-f003:**
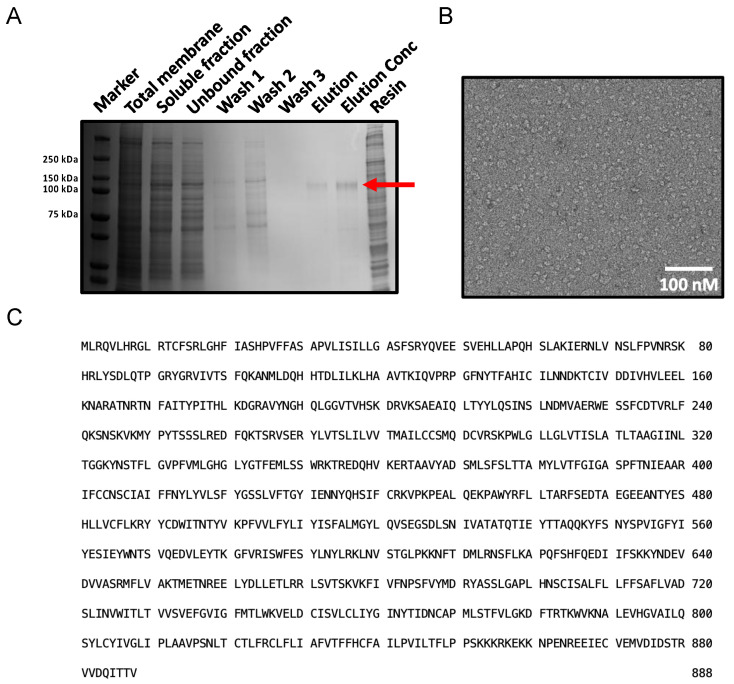
Optimised PTCHD1 purification. (**A**) SDS-PAGE shows a clear band corresponding to the 101 kDa size of PTCHD1. (**B**) A representative EM image of negatively stained, purified PTCHD1 at 25k magnification shows evenly distributed homogeneous particles. (**C**) Highlighted residues of PTCHD1 were identified by mass spectrometry following purification.

**Figure 4 ijms-24-02682-f004:**
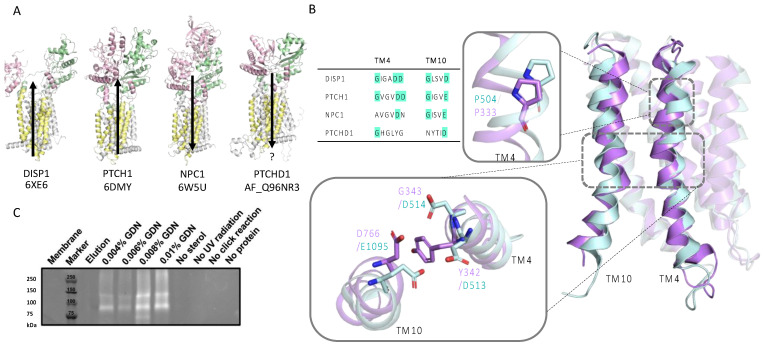
PTCHD1 cholesterol binding. (**A**) Patched domain-like family cholesterol transporters contain an SSD (yellow) and two ECDs (pink and green). Direction of cholesterol transport with respect to the concentration gradient is indicated with black arrows. The direction in which PTCHD1 transports cholesterol has not been experimentally verified (indicated with “?” under the arrow). (**B**) A sequence alignment of cholesterol transporters showing DISP1 and PTCH1 have conserved RND motifs. NPC1 only has one conserved motif, while PTCHD1 has neither. Structural alignment of PTCHD1 (purple) and PTCH1 (blue) SSDs demonstrates high structural similarity. The top inset shows SSDs shows a conserved, functionally critical, proline on TM4. The bottom inset shows the charge neutralising residues at the TM4:TM10 interface of PTCH1 and corresponding residues of PTCHD1. (**C**) SDS-PAGE of PhotoClick cholesterol assay. Clear bands are visible in each lane of PTCHD1 with GDN around the estimated 130 and 100 kDa molecular weight of glycosylated and unglycosylated PTCHD1, but not in control sample lanes.

**Figure 5 ijms-24-02682-f005:**
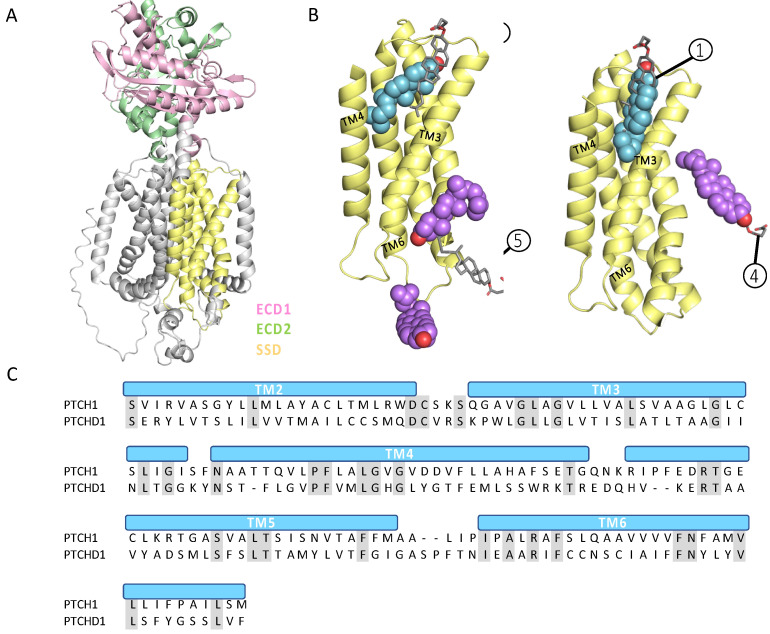
Cholesterol docking to PTCHD1. (**A**) PTCHD1 (AF-Q96NR3) consists of two ECDs (pink and green), and 12 transmembrane helices, five of which form the SSD (yellow). (**B**) Cholesterols docked to PTCHD1 SSD (left, AF-Q96NR3) aligning well with PTCH1 (6RMG) cholesterol-binding sites 1 and 5 (superposed PTCH1 cholesterols in grey). Cholesterols docked to the AlphaFold2-predicted PTCH1 SSD (right, AF-Q13635), right, and align with experimentally determined cholesterol-binding sites 1 and 4 (superposed PTCH1 (6RMG) cholesterols in grey). (**C**) A sequence alignment of PTCH1 and PTCHD1 SSDs.

**Figure 6 ijms-24-02682-f006:**
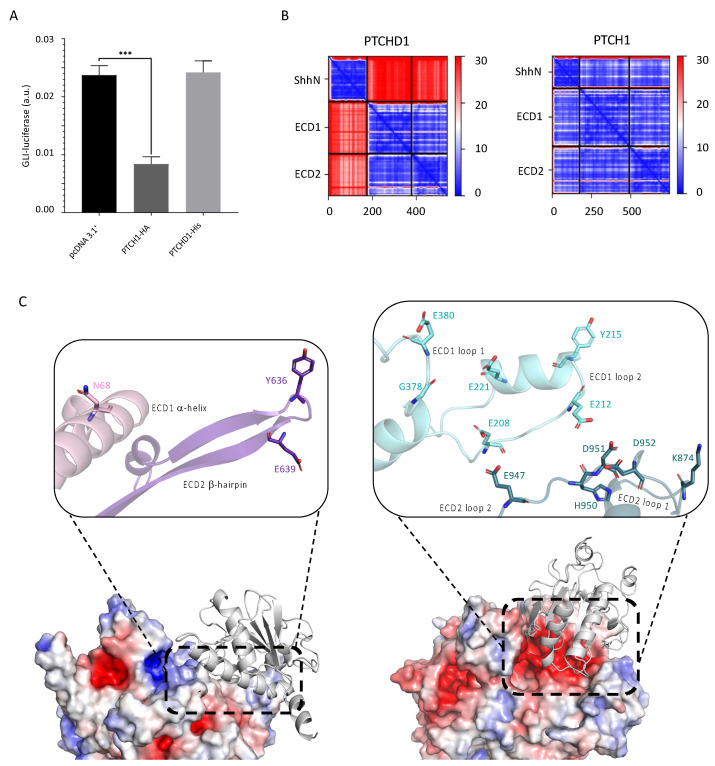
PTCHD1 does not inhibit Hh pathway activation nor bind ShhN. (**A**) A GLI-luciferase assay of *Ptch1*^−/−^ MEFs shows transient transfection with PTCH1-HA leads to a significant (65%) inhibition of the Hh pathway (*** *p* < 0.001), while the level of GLI-luciferase expression remains unchanged in cells transfected with PTCHD1-His or empty plasmid (pcDNA3.1^+^). (**B**) Figures depicting the predicted aligned error between residues show high error rate (red) in predicted complex formation between ShhN and PTCHD1 ECDs, while the estimated error is low (blue) between the two ECDs. On the other hand, the predicted error is low between ShhN and PTCH1 ECDs. (**C**) AlphaFold2 predicted complexes of PTCHD1 ECD1:ECD2:ShhN (left) and PTCH1 ECD1:ECD2:ShhN (right). Insets show close-ups of the secondary structures and hydrogen-bond forming residues involved in ShhN binding.

**Figure 7 ijms-24-02682-f007:**
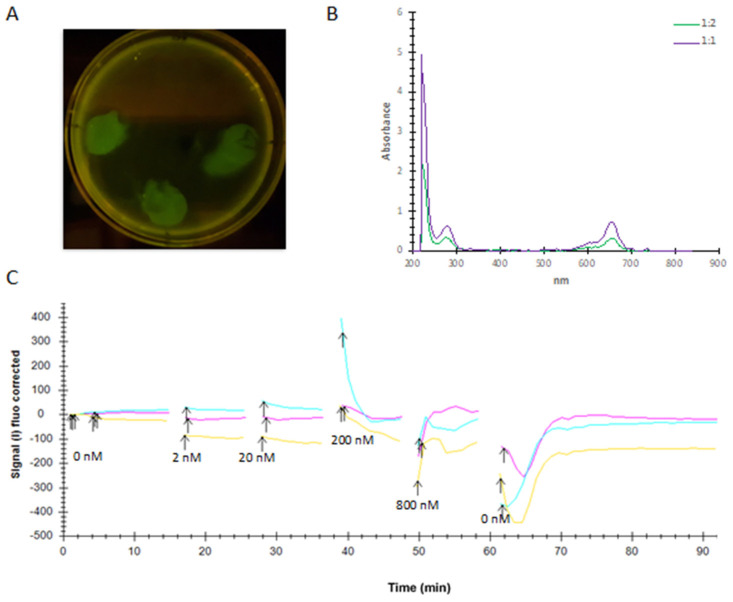
LigandTracer experiment. (**A**) Fluorescence of tethered Sf9 cells confirms expression of pFastBac-PTCHD1-CVGH on three distinct spots on the Petri dish. The control spot (top) shows no fluorescence. (**B**) Absorbance trace of 1:1 and 1:2 ShhN labelled with DyLight^TM^ 650. Clear absorbance peaks can be seen at 220, 280, and 650 nm. (**C**) Triplicate ligand binding (15–60 min) and dissociation (60–90 min) traces. Signal intensity has been normalized to the background signal (i.e., signal trace of control cells). Arrows indicate changes in ShhN concentration.

**Figure 8 ijms-24-02682-f008:**
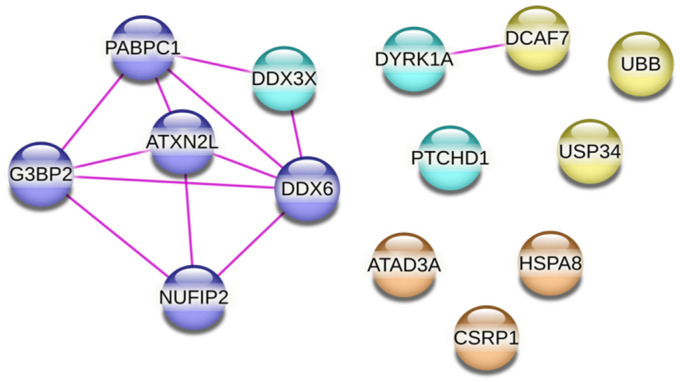
STRING analysis of PTCHD1 -specific interactors. Pink lines show experimentally determined physical interactions between PTCHD1 interactors identified through STRING analysis. DDX3X, PABPC1, ATXN2L, DDX6, G3BP2, and NUFIP2 form a cluster of physical interactions. Proteins in cyan are associated with autism. Others are coloured by function: purple—RNA metabolism, yellow—ubiquitylation, and orange—miscellaneous.

**Figure 9 ijms-24-02682-f009:**
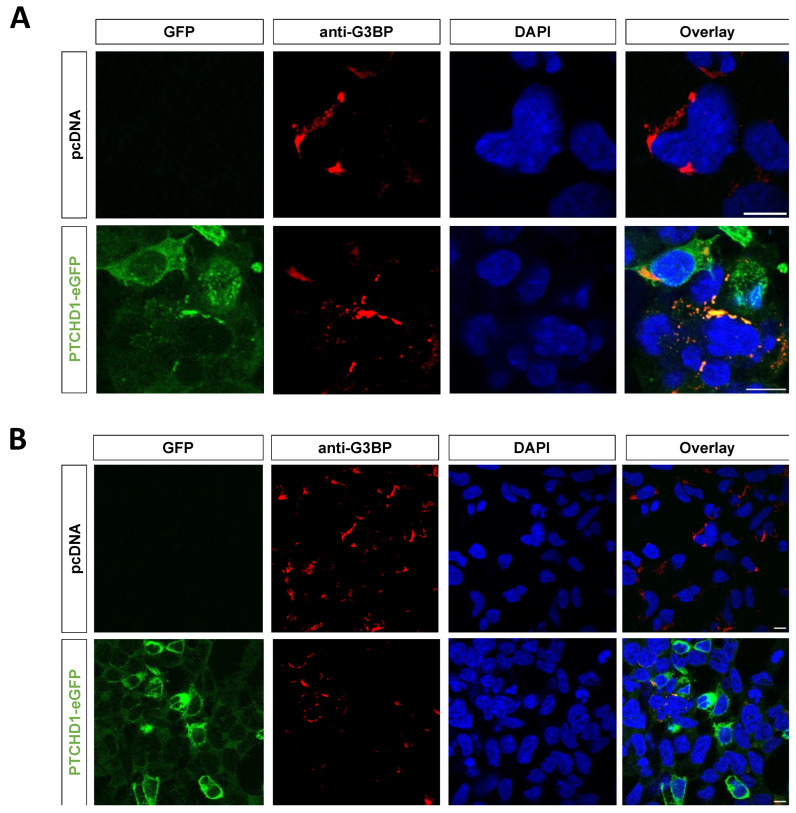
PTCHD1 is associated with stress granule markers. PTCHD1-eGFP or empty pcDNA vector was transfected into HEK293 cells. After 24 h, cells were fixed and stained with anti-G3BP Ab and AlexaFluor564-conjugated secondary antibody and mounted with DAPI. Scale bar = 10 μm, showing a magnified view of selected areas (**A**) and the field view (**B**) with an oil-immersion objective.

**Table 1 ijms-24-02682-t001:** PTCHD1-specific interactors.

Protein Name	No Unique Peptides	Reported Function
ATXN2L *	39	Regulation of stress granule and P-body formation.
NUFIP2 *	25	RNA binding protein found in stress granules.
PABP1 *	23	Binds the poly(A) tail of mRNA, regulates processes of mRNA metabolism such as pre-mRNA splicing and mRNA stability.
HSPA8 *	12	Molecular chaperone implicated in a wide variety of cellular processes, including protection of the proteome from stress, and acts as a repressor of transcriptional activation.
DCAF7 *	10	Associates with DIAPH1 and controls GLI1 transcriptional activity. Binds DYRK1A through its CTD. Inhibits DYRK1A dependent GLI activation and nuclear retention [26].
DYRK1A ^A^	16	Dual-specificity kinase which possesses both serine/threonine and tyrosine kinase activities. Plays an important role in double-strand break (DSB) repair following DNA damage. Phosphorylates GLI1 to promote nuclear localisation and transcriptional activity [27].
DDX3X *^A^	6	ATPase/helicase activity. Required for efficient stress granule assembly via interaction with eIF4E. Direct interaction partner of PABP1.
UBB *	4	Involved in protein synthesis and degradation. Highly expressed in gonadotropin-releasing hormone neurons [26].
CSRP1 *	4	Potential role in neuronal development. Cysteine-rich domain is highly conserved in steroid receptors.
G3BP2 *	4	Scaffold protein with essential role in cytoplasmic stress granule formation.
ATAD3A *°	8	Required for enhanced channelling of cholesterol for hormone- dependent steroidogenesis.
UBP34	4	Ubiquitin hydrolase that removes conjugated ubiquitin from AXIN1 and AXIN2, regulating Wnt signalling pathway downstream of the beta-catenin destruction complex.
DDX6 *	4	Essential for the formation of P-bodies and ribonucleoprotein particles.

Unless otherwise indicated, reported functions were derived from UniProtKB entries. ^A^ associated with autism. * Corroborated by [6]. ° Also identified as an interaction partner in PTCH2-FLAG IP Mass Spectrometry.

**Table 2 ijms-24-02682-t002:** Statistical overrepresentation among PTCHD1-specific interactors.

Cellular Component			Fold Enrichment	*p*-Value	False Discovery Rate
	cytoplasmic stress granule	(GO:0010494)	>100	10 × 10^−12^	2 × 10^−8^
	cytoplasmic ribonucleoprotein particle	(GO:0036464)	38	5 × 10^−9^	5 × 10^−6^
	ribonucleoprotein particle	(GO:0035770)	36	8 × 10^−9^	5 × 10^−6^
	ribonucleoprotein complex	(GO:1990904)	14	2 × 10^−6^	8 × 10^−4^
	supramolecular complex	(GO:0099080)	9	4 × 10^−7^	2 × 10^−4^
	intracellular non-membrane-bounded organelle	(GO:0043232)	3	1 × 10^−4^	5 × 10^−2^
	non-membrane-bounded organelle	(GO:0043228)	3	1 × 10^−4^	4 × 10^−2^
**Biological process**					
	stress granule assembly	(GO:0034063)	>100	2 × 10^−9^	3 × 10^−5^
	non-membrane-bound organelle assembly	(GO:0140694)	25	10 × 10^−7^	8 × 10^−3^
**Molecular function**					
	RNA binding	(GO:0003723)	8	2 × 10^−6^	8 × 10^−3^

## Data Availability

The original contributions presented in the study are included in the article and Appendix A. Further inquiries can be directed to the corresponding author.

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
