# Peer review of "PTCHD1 Binds Cholesterol but Not Sonic Hedgehog, Suggesting a Distinct Cellular Function"

_ijms, 2023, doi:10.3390/ijms24032682_

Round 1

Reviewer 1 Report (New Reviewer)

Manuscript ID: ijms-2030948

Title: PTCHD1 binds cholesterol but not Sonic Hedgehog, suggesting a distinct cellular function

In this research, the authors studied the potential functions of PTCHD1 via in vitro and in silicon studies. They proposed that different from PTCH1 and NPC1, PTCHD1 binds cholesterol but not sonic hedgehog and cannot inhibit canonical hedgehog signaling. The authors proposed that PTCHD1 might play a role in stress granule formation based on the identification of PTCHD1-interactors.

Specific comments:

1.       The authors proposed that PTCHD1 might play a role in stress granule formation based on the identification of PTCHD1-interactors. However, PTCHD1 is a membrane protein with localization on the plasma membrane, whereas stress granules are cytosolic membraneless organelles. It is better to verify this hypothesis by subcellular localization analysis via confocal or other methods to check whether PTCHD1 has distributions in the stress granules, and whether PTCHD1 colocalize with the stress granule components such as G3BP and/or DDX6.

2.       Figure 1C, what is the meaning of the label GFP on the right of the gel.

3.       Figure 2a and 4c, please add the molecular weights of the markers.

4.       Figure 7C, the label of the horizonal axis, please use “Time (min)” to keep consistent with those in Figure 2.

Author Response

We are glad to report that we were able to address all of the reviewer’s comments in our revised manuscript. Specifically:

  1. The authors proposed that PTCHD1 might play a role in stress granule formation based on the identification of PTCHD1-interactors. However, PTCHD1 is a membrane protein with localization on the plasma membrane, whereas stress granules are cytosolic membraneless organelles. It is better to verify this hypothesis by subcellular localization analysis via confocal or other methods to check whether PTCHD1 has distributions in the stress granules, and whether PTCHD1 colocalize with the stress granule components such as G3BP and/or DDX6.”

We have now incorporated confocal imaging data that confirms co-localisation of a fraction of PTCHD1 with the stress granule marker G3BP in perinuclear regions (new Figure 9). It has been previously reported that stress granules containing ER-targeted mRNAs are associated with ER membranes (https://rnajournal.cshlp.org/content/27/10/1241.long). Given that PTCHD1 is synthesised in the ER, it is similarly plausible that its cytosolic domains interact with stress granule proteins, as suggested by our imaging data.

  1. Figure 1C, what is the meaning of the label GFP on the right of the gel.

Figure 1C shows the “in gel” GFP fluorescence of the indicated purification fractions using blue light, a common method to detect abundant recombinant GFP-tagged proteins in SDS-PAGE gels. It is described in Methods 4.2 section.

  1. Figure 2a and 4c, please add the molecular weights of the markers.

Our apologies, the MW have now been added.

  1. Figure 7C, the label of the horizontal axis, please use “Time (min)” to keep consistent with those in Figure 2.

We have replaced minutes by “min”, as requested.

Reviewer 2 Report (New Reviewer)

The authors reported PTCHD1, unlike PTCH1, did not bind to Sonic hedgehog but cholesterol. The authors presented ample evidence to support this conclusion. However, this conclusion is reported already in multiple previous studies. For example, directly quoting Ung et al. Molecular Psychiatry 2018 "We also report that PTCHD1 is unable to rescue the canonical sonic hedgehog (SHH) pathway in cells depleted of PTCH1, suggesting that both proteins are involved in distinct cellular signalling pathways." The fact that PTCHD1 did not bind Shh was also reported by Tora et al. J Neurosci 2017, Pastore et al. Genes 2022. The structure of PTCHD1 was predicted by Alphafold which is an open source web based tool. While I do not intend to undermine the findings of the authors, the results from this study  are far from providing sufficient new results to the field. For example, one would expect the authors to report what cholesterol (not if) PTCHD1 would bind, or how the cholesterol binding affinity varies between the PTCH family proteins. Simply put, this study is a replica of previous findings. The positive is the data quality looks solid. However, I would recommend rejection of the manuscript, unless more new experiments are performed to provide new insights into the function of PTCHD1.

Author Response

We divided the answer to the reviewer’s comment by parts, as follows:

  1. “The authors reported PTCHD1, unlike PTCH1, did not bind to Sonic hedgehog but cholesterol. The authors presented ample evidence to support this conclusion.”

We thank the reviewer for their confirmation that the conclusions are strongly supported by the data.

  1. “However, this conclusion is reported already in multiple previous studies. For example, directly quoting Ung et al. Molecular Psychiatry 2018 "We also report that PTCHD1 is unable to rescue the canonical sonic hedgehog (SHH) pathway in cells depleted of PTCH1, suggesting that both proteins are involved in distinct cellular signalling pathways."

We are aware of the study by Ung et al., which is cited in our manuscript. However, other studies reported that PTCHD1 can mimic PTCH1 (Noor et al.; Chung et al.). The value of our conclusion in this regard, which agrees with Ung et al., is solving an existing controversy. The Riobo-Del Galdo lab has over 20 years of experience in Hedgehog signalling and has carefully compared the activity of PTCHD1 and PTCH1 in Ptch1-deficient cells to reach this conclusion. Thus, although it is not the first time this aspect is reported, we believe it is very valuable to provide a definitive answer to this question.

  1. The fact that PTCHD1 did not bind Shh was also reported by Tora et al. J Neurosci 2017, Pastore et al. Genes 2022.”

We agree that this aspect is not totally novel. However, as opposed to Tora et al. who measured binding of a Shh-Fc fragment to whole cells, we use intact Shh and purified PTCHD1 protein. The use of intact Shh precludes any potential steric clash of the Fc fragment with PTCHD1 that could have resulted in an artificial negative binding result in Tora’s study. Furthermore, we support the lack of Shh binding with analysis with in silico analysis of the predicted interaction surfaces (Figure 6 c).

The paper of Pastore et al. is a review article that cites Tora’s finding.

  1. “The structure of PTCHD1 was predicted by Alphafold which is an open source web based tool. While I do not intend to undermine the findings of the authors, the results from this study  are far from providing sufficient new results to the field. For example, one would expect the authors to report what cholesterol (not if) PTCHD1 would bind, or how the cholesterol binding affinity varies between the PTCH family proteins. Simply put, this study is a replica of previous findings. The positive is the data quality looks solid.”

Our docking analysis -completely novel for PTCHD1- show the potential cholesterol binding sites and the free energies of binding. This analysis was performed on the predicted AlphaFold2 structure of the sterol sensing domain of PTCHD1, which had a very high predictive confidence score, in comparison with the highly similar SSD of PTCH1 which has been solved by several groups. We further confirmed the existence of cholesterol binding using a click chemistry assay using purified PTCHD1 (Figure 4). We believe the reviewer asks if cholesterol or an oxysterol derivative is the main binder. We would like to point out that even in the case of the solved cryo-EM structures of PTCH1, the exact nature of the sterols bound to it is unclear. In addition, the reviewer might have missed that we compared the binding affinity of cholesterol to PTCHD1 and PTCH1 in our docking analysis (section 2.2.3 and supplementary Table 1).

  1. “However, I would recommend rejection of the manuscript, unless more new experiments are performed to provide new insights into the function of PTCHD1.”

We regret this reviewer did not see the additional value presented by our study. We have:

  • Developed an experimental strategy to purify PTCHD1 (novel)
  • Solved a controversy around the activity of PTCHD1 as a Hh pathway repressor
  • Ruled out any interaction with Shh using purified intact proteins (novel)
  • Demonstrated that PTCHD1 binds cholesterol (novel)
  • Identified through extensive in silico analysis the potential cholesterol binding sites and compared their affinities to the comparable sites in PTCH1 (novel)
  • Identified a number of PTCHD1-specific interacting proteins which do NOT interact with PTCH1 or PTCH2 (novel comparison and identification of few additional proteins never reported)
  • Confirm that a fraction of PTCHD1 associates with stress granules (novel and NEW Figure 9)

We believe that these findings, both the novel and the confirmatory, represent a valuable contribution to the understanding of PTCHD1 biology and will be instrumental in the next steps to uncover its role in neurodevelopmental disorders.

Round 2

Reviewer 1 Report (New Reviewer)

The authors have addressed most of my comments. However, there are several issues needing to be clarified before acceptance:

1, As for the label "GFP" in the figures, it is better to specify the GFP-fused protein(s) but not GFP since GFP-fused proteins and GFP are two different things.

2. The authors have conducted colocalization analysis using confocal imaging. Generally, stress granules are appeared under stressed conditions but rarely under normal culturing conditions.  I noticed that the authors did not treat the cells using stressed conditions. Does that mean that stress granules were formed under normal cultivating conditions? Please explain it or clarify the conditions used for stress granule formation.

Author Response

We have performed the minor changes indicated by the reviewer:

1- replaced the GFP label next to the blot in Fig 2 by PTCHD1-eGFP

2- explained in the text that low level of stress granules were observed in transiently transfected cells without additional stressor.

We thank this reviewer for these suggestions that improved our manuscript.

This manuscript is a resubmission of an earlier submission. The following is a list of the peer review reports and author responses from that submission.

Round 1

Reviewer 1 Report

Hiltunen and colleagues report on a molecular and functional study of the PTCHD1 protein, mutations of which have been described in patients with neurodevelopmental disorders.

To understand the biochemical function and structural organization of PTCHD1, the authors developed a method for the production of PTCHD1 in eukaryotic insect cells and a purification method of PTCHD1-containing membrane extracts.  They also performed various in vitro and in silico experiments indicating that PTCHD1 does not function in the sonic hedgehog canonical pathway, unlike its homolog PTCH1, but it can bind cholesterol similarly to PTCH1. Lastly, the authors performed PTCHD1 protein interactions analysis and they detected a list of interactors involved in cell stress response and in RNA granule formation.  

Overall, I find this work interesting and it would provide original findings and novel clues related to the function of PTCHD1 as a protein displaying a distinct role from PTCH1. The production and purification optimizations of PTCHD1 from insect cells is original. 

Conversely, the role of PTCHD1 in the regulation of the canonical hedgehog signalling has been already described in at least 2 publications using the same approach (i.e. Ptch1-/- MEFs). The last part of their manuscript describes PTCHD1 interactors characterized by co-immunoprecipitation experiments followed by mass spectrometry, but without confirmation using western blot analyses.

Please find below my major comments:

. The materials and methods section must be significantly improved: 

The cloning and production of PTCHD1 must include the information about the cloning procedures, the RefSeq of PTCHD1 cDNA is not specified.

The Western blotting experiments are not appropriately described: What is the amount of protein lysates loaded in SDS-PAGE? Were the protein lysates heat-denaturated before gel-loading etc…? It is crucial to indicate as many information as necessary since the authors claim in their manuscript that they developed a method for expressing and purifying PTCHD1.

. The authors indicate in Figure 1 that the 100 kDa band in the blot image (panel C) corresponds to PTCHD1. Since this membrane has been incubated with a GFP antibody, the band should label the PTCHD1-GFP protein (according to the plasmid construct in panel A), which will be detected at 125-127 kDa. The authors must comment on that point, as it is essential for the study to confirm that this is the full-length wild-type protein that has been produced, and not a truncated form. Furthermore, they need to clarify more precisely in the text the type of PTCHD1 protein that is detected in the blots displayed in the various figures (Figs 1, 2, 3, 4): GFP-tagged, untagged etc..

. The authors must include a paragraph on the statistical analysis of their experiments, particularly to indicate how they performed their analysis in Figure 6A.

. Figure 2B: how many replicates have been performed? If the graph shows only the date from one experiment, it is not methodologically appropriate. Please provide at least 2-3 replicates to confirm your solubilization and purification process strategy.

. Figure 4C: The authors indicate that the two bands correspond to the 100 kDa unglycosylated and 130kDa glycosylated forms of PTCHD1. They need to confirm their hypothesis using endoglycosidase assays. Furthermore, the “100kDa” band is rather positioned between 75 and 100 kDa, according to the blot image in Fig 4C.   

. I have several comments on the co-immunoprecipitation experiments:

The authors actually identified proteins partners of PTCHD1 using HEK293 cells transfected with PTCHD1-CVGH (including GFP and His Tags at the C-terminus of PTCHD1). 

First, they must confirm their protein interaction results using western blot analyses of their Co-IP samples. 

Second, the use of PTCH2 as a control to exclude non-specific PTCHD1 protein interactions is not clearly justified.

Third, given the neuronal function of PTCHD1, I don’t understand why the authors did not use neuronal cell lines (neuro2a for example). Furthermore, it would have been probably more informative to use brain lysates as prey, since it appears from their experiment that they actually analyzed the endogenous HEK293 proteome.

Finally, the GFP-His tag positioned at the C-terminus of PTCHD1 prevent the PDZ-binding motif to be functional. Have the authors tried to insert the His tag in other parts of the PTHCD1 protein sequence?    

Minor comments:

. Please respect the HUGO Gene/protein nomenclature in the 4.1 section : PTCHD1 for the human cDNA, and PTCHD1 for the human protein

. References 9 and 26 are identical; it seems that reference 26 is not included in the text.

Reviewer 2 Report

The manuscript entitled “PTCHD1 binds cholesterol but not Sonic Hedgehog, suggesting a distinct cellular function” is interesting piece of scientific advancement. However, there are concerns on the study.

The study utilised only HEK293 cells which is known for the overexpression of PTCH1 throughout the plasma membrane of HEK293T cells, is there any similar observation on PTCHD1. Any observation on the endogenously expression of PTCHD1. How authors confirm the distinct cellular function using only one cell line.

Authors have stopped with the Docking analysis of cholesterol to the putative PTCHD1. This study needs to be completed with the support of molecular dynamics simulation.

Minor concern is the title not reflecting the observation.

Objective of the study is not defined. I am wondering about the statement “Understanding if PTCHD1 could have a similar function could provide key information on the association of PTCHD1 with ASD.” Authors have missed in most of the places to state the aim.

Round 2

Reviewer 2 Report

The major concern still remains ... How authors suggest the distinct cellular function using only one cell line.